# Monitoring real-time transmission heterogeneity from incidence data

**Yunjun Zhang**[1,2]*, **Tom Britton**[3], **Xiaohua Zhou**[1,2,4,5]*

**1** Department of Biostatistics, School of Public Health, Peking University, Beijing, China, **2** Center for Statistical Science, Peking University, Beijing, China, **3** Department of Mathematics, Stockholm University, Stockholm, Sweden, **4** Beijing International Center for Mathematical Research, Peking University, Beijing, China, **5** School of Mathematical Sciences, Peking University, Beijing, China

* yunjun.zhang@pku.edu.cn (YZ); azhou@math.pku.edu.cn (XZ)

**Data Availability Statement:** All code and data are available at https://github.com/yunPKU/infer_heterogeneity_from_incidence.

**Funding:** Y.J.Z and X.H.Z acknowledge support from National Natural Science Foundation of China

## Abstract

The transmission heterogeneity of an epidemic is associated with a complex mixture of host, pathogen and environmental factors. And it may indicate superspreading events to reduce the efficiency of population-level control measures and to sustain the epidemic over a larger scale and a longer duration. Methods have been proposed to identify significant transmission heterogeneity in historic epidemics based on several data sources, such as contact history, viral genomes and spatial information, which may not be available, and more importantly ignore the temporal trend of transmission heterogeneity. Here we attempted to establish a convenient method to estimate real-time heterogeneity over an epidemic. Within the branching process framework, we introduced an instant-individualheterogenous infectiousness model to jointly characterize the variation in infectiousness both between individuals and among different times. With this model, we could simultaneously estimate the transmission heterogeneity and the reproduction number from incidence time series. We validated the model with data of both simulated and real outbreaks. Our estimates of the overall and real-time heterogeneities of the six epidemics were consistent with those presented in the literature. Additionally, our model is robust to the ubiquitous bias of under-reporting and misspecification of serial interval. By analyzing recent data from South Africa, we found evidence that the Omicron might be of more significant transmission heterogeneity than Delta. Our model based on incidence data was proved to be reliable in estimating the real-time transmission heterogeneity.

## Author summary

The transmission of many infectious diseases is usually heterogeneous in time and space. Such transmission heterogeneity may indicate superspreading events (where some infected individuals transmit to disproportionately more susceptibles than others), reduce the efficiency of the population-level control measures, and sustain the epidemic over a larger scale and a longer duration. Classical methods of monitoring epidemic spread centered on the reproduction number which represent the average transmission potential of the epidemic at the population level, but failed to reflect the systematic variation in

(Grant number: 82041023), the Bill & Melinda Gates Foundation (Grant number: INV-016826). T. B. is supported by The Swedish Research Council (grant 2020-04744). The funders had no role in study design, data collection and analysis, decision to publish, or preparation of the manuscript.

**Competing interests:** No authors have competing interests.

transmission. Several recent methods have been proposed to identify significant transmission heterogeneity in the epidemics such as Ebola, MERS, COVID-19. However, these methods are developed based on some sophisticated information such as contact history, viral genome and spatial information, of the confirmed cases, which are typically field-specific and not easy to generalize. In this study, we proposed a simple and generic method of estimating transmission heterogeneity from incidence time series, which provided consistent estimation of heterogeneity with those records with detailed data. It also helps in exploring the transmission heterogeneity of the newly emerging variant of Omicron. Our model enhances current understanding of epidemic dynamics, and highlight the potential importance of targeted control measures.

## Introduction

The transmission of infectious disease is typically uneven or heterogeneous in terms of time and space due to a complex mixture of host, pathogen and environmental factors [1–6]. High level of transmission heterogeneity may indicate superspreading events (SSEs) in which certain individuals infect a greater large number of secondary cases than average [1], invoking the so-called 20–80 rule. It has been documented that the SSEs considerably reduced the efficiency of population-level control measures [1] and played a key role in dramatically driving the spread of many pathogens in scale and duration, including severe acute respiratory syndrome (SARS) [7], Middle East Respiratory Syndrome (MERS) [8], Ebola [3, 4] and COVID-19 [6, 9, 10]. Therefore, monitoring the degree of transmission heterogeneity and its change could be vital for epidemic forecasting and efficient intervention in infectious disease epidemiology.

Mathematically, the transmission heterogeneity is represented by the variation in offspring distribution, namely, the distribution of secondary cases that may be generated by a given infectious case. Classical methods of estimating heterogeneity rely heavily on reconstructing the offspring distribution. As the epidemiological links among reported cases are complex, this reconstruction poses considerable challenges in both data collection and model building. According to different types of data used in the reconstruction, the existing methods of inferring heterogeneity can be grouped into three categories. The first category are methods based on contact-tracing-data. By interviewing patients to document their contacts with other infected patients, all or most of the cases could be positioned in the network of transmission, and the resulting empirical offspring distribution could be directly used to estimate the transmission heterogeneity [1, 4, 10, 11].

The second category is based on virus-sequence-data. For many pathogens, in particular RNA viruses, the evolution of virus genomes is almost at the same rate as the transmission of the disease, which makes it possible to extract epidemiological information from genetic analysis [12, 13]. Many studies showed that the virus phylogeny reconstructed from the virus sequence sampled from the infected individuals reflected the underlying transmission history of the epidemic, with the branching events in a phylogeny corresponding to transmission events in the past. By incorporating the level of heterogeneity into the likelihood function of the virus phylogeny, it is possible to estimate the heterogeneity as well as other epidemiological parameters from the sampled sequence data [2, 14, 15].

For the third category, individual-level spatial information has been integrated to reconstruct the transmission history in recent years. By developing a continuous-time spatiotemporal transmission model with a distance-based kernel to characterized the infectiousness between individuals as a function of the mutual distance, it is possible to infer explicitly the

mean offspring distribution of each case and hence to infer the transmission heterogeneity and other epidemiological parameters [3, 9, 16].

Although considerable progress has been made for analyzing heterogeneity, these methods also showed some theoretical and practical limitations. Firstly, all these methods required context-specific information which could be hard to obtain and/or could be erroneous. For example, the contact tracing in epidemiological investigation may be time-consuming and subjective [17] and has to be limited to a certain number of infected cases. In viral genetic analysis, the commonly used correspondence between the reconstructed viral phylogeny and the transmission history may be biased if there are within-host evolution and recombination in viral genomes [18]. When incorporating the spatial information, the model simply assumes that transmission occurred mostly within close residence because of the lack of detailed individual movement data, which is only appropriate under certain control measures [3, 9].

In addition, most of existing studies assumed a constant level of heterogeneity for an epidemic under study, which may in fact grow and/or decline through the epidemic. This simplification would bring some computational benefit but failed to characterize the temporal change of heterogeneity over the epidemic. Although Lau et al [3, 9] compared the degree of heterogeneity in different periods of an outbreak (i.e., before and after deploying the control measures), it could still be hard to reflect the real-time development of the epidemic and consequently lead to inadequacy in epidemic control to a certain extent.

Monitoring real-time transmission dynamics from incidence data has drawn a lot of research efforts. Several tools for the estimating of real-time reproduction number based on incidence data had been developed with successful applications [19–21], but the study on real-time transmission heterogeneity is so far rather limited. In some recent studies, researchers suggested the relationship between the transmission heterogeneity and the incidence over an epidemic [22–25], but none have attempted to accurately delineate the heterogeneity with incidence data and to compare with those records in literatures. In this study, we attempted to develop a simple method to estimate the transmission heterogeneity on the basis of incidence data. Specifically, we extended the homogeneous transmission model in [19, 20] to allow for the variation of infectiousness at different times and among different people, and consequently generated real-time estimates of transmission heterogeneity and reproduction number simultaneously. Moreover, we evaluated this model with both simulated data and historic epidemic data, which turned out to be consistent with that of those involving contact-tracing or spatial data. Our model performed robust even in the presence of measurement errors such as under-reporting or misspecification of serial interval. We further explored the transmission heterogeneity of the new SARS-CoV-2 variant Omicron based on the incidence time series from South Africa.

## Materials and methods

### Renewal process model of transmission

We considered an outbreak observed regularly (in days, weeks or months) over the time period $1 \leq t \leq T$. Let $I_t$ be the incidence or number of newly infected cases at time $t$, and the epidemic curve till time $t$ is denoted as $\bar{I}_1^t = \{I_1, I_2, \cdots, I_t\}$. For simplicity, we excluded the possibility of imported case during the study period. However, this restriction could be relaxed by discriminating the effect on newly infections of local/imported cases as in [20].

We adopted the renewal process to model the transmission of the infectious disease. Under the standard renewal process model [19], the newly infected at time $t$ (i.e., $I_t$) is generated by all the infectious individuals who had been infected before time $t$ according to a Poisson

relation as:

$$I_t | \bar{I}_1^{t-1} \sim \text{Pois}(R_t \Lambda_t) \tag{1}$$

where "|" stands for conditions, Pois stands for *Poisson* distribution, and $\bar{I}_1^{t-1}$ represents the incidence data between time 1 and $t-1$. The parameter of $R_t$ is the instantaneous reproduction number, representing the average number of secondary cases that caused by a random case at time $t$ if circumstances remained the same after that [19, 26]. The quantity $\Lambda_t = \sum_{s=1}^{t-1} I_s w_{t-s}$, known as the total infectiousness, characterizes how many past effective cases contribute to the newly observed case-count at time $t$. The weight $w_{t-s}$ defines the impact of each past case on the newly infection, which could be approximated by the generation time distribution or the serial interval distribution.

## Instant-individual reproduction number

In this study, we aimed to extend the standard model to allow for transmission heterogeneity during the transmission process. To characterize the effect of each infected individual on new infection at a particular time point, we introduce the "instant-individual reproduction number" (IIRN), denoted as $v_{s,t}^i$, representing the expected number of secondary cases generated at time $t$ by the $i$-th individual infected at time $s$ (where $s < t$). We also use the Poisson distribution to model the stochastic effect in transmission [1], so the number of secondary cases caused by a particular case (i.e., offspring distribution) in the given context is $\text{Pois}(v_{s,t}^i)$. In addition, we adopted the assumption that the offspring distributions of different cases were independent, so the incidence $I_t$ is the sum of these *Poisson*-distributed variables. In other words, $I_t$ is *Poisson*-distributed with the composite rate of $v_t = \sum_{s<t} \sum_i v_{t,s}^i$.

The concept of IIRN provides a new tool to explore the variation of infectiousness between different individuals and among different times. Next we study how the standard renewal process model and two recently proposed heterogenous transmission models fit within this framework. The standard renewal process model is a homogeneous transmission model, which assumed a constant IIRN for all the infected cases who had been infected at the same time. Since $\Lambda_t = \sum_{s \le t} w_{t-s} I_s = \sum_{s \le t} \sum_{i=1}^{I_s} w_{t-s}$, the standard model (1) is identical to assume $v_{s,t}^i = w_{t-s} R_t$ which has the composite rate at time $t$ as $v_t = \sum_{s \le t} \sum_i v_{t,s}^i = R_t \Lambda_t$. This model, while useful for monitoring the average transmission potential, fails to account for the variation in infectiousness particular found in the those superspreading events.

Another common method of allowing for transmission heterogeneity is an instant-level heterogeneity model [22, 25]. This model extended the standard model (1) by replacing the instantaneous reproduction number $R_t$ with an instant-related random variable $\eta_t$ for all the infected cases, that is,

$$v_{t,s}^i = w_{t-s} \eta_t, \text{ where } \eta_t \sim \Gamma(k_t, \frac{k_t}{R_t}).$$

where $\Gamma(\cdot, \cdot)$ stands for *Gamma* distribution in the shape-rate parameterizations. Therefore, the composite rate under this model is $v_t = \sum_{s \le t} \sum_i v_{t,s}^i = \Lambda_t \eta_t$, which is weighted gamma-distributed conditioned on the incidence curve $\bar{I}_1^{t-1}$ as $v_t | \bar{I}_1^{t-1} \sim \Gamma(k_t, \frac{k_t}{\Lambda_t R_t})$. And the incidence $I_t$ is Negative Binomial distribution as (NegB indicating *Negative Binomial distribution*):

$$I_t | \bar{I}_1^{t-1} \sim \text{NegB}(k_t, \frac{k_t}{\Lambda_t R_t + k_t})$$

This model accounted for the variation in infectiousness at different times, which could be useful in epidemic forecasting in the long term [22, 25]. But this model overlooked the variation in infectiousness of different infectious individuals, and hence failed to identify the exact degree of heterogeneity from incidence data (showed in Results).

Recently, Johnson et al [27] proposed an individual-level heterogeneity model to characterize transmission heterogeneity within the renewal process framework. For a particular case (e.g., $i$-th case) infected as time $s$, the model assumed its infectiousness at time $t$ as $v_{t,s}^i = w_{t-s}\eta_t^i$, where $\eta_t^i$ is randomly drawn (per case) as $\eta_t^i \sim \Gamma(k_t, \frac{k_t}{R_t})$.

With this model, the composite rate of newly infection at time $t$ is $v_t = \sum_{s \leq t}\sum_i v_{t,s}^i = \sum_s w_{t-s}\Theta_s$, where $\Theta_s = \sum_i \eta_s^i \sim \Gamma(k_s * I_s, \frac{k_s}{R_s})$, and was referred to as the disease momentum [27], representing the total infectiousness of all the cases infected at time $s$. As the weighted summary of *Gamma* variables is not *Gamma* distributed, the incidence $I_t$ can only be approximated by

$$I_t | \bar{I}_1^{t-1} \sim \text{Pois}(\sum_s w_{t-s}\Theta_s).$$

This model characterized the individual level transmission heterogeneity at the cost of introducing a large number of nuisance parameters of disease momentums $\{\Theta_s\}$. These nuisance parameters destroyed the conditional independence of incidence data among different times, and incurred considerable computational complexity in the analysis of incidence time series, which hinder the accuracy of estimating parameters of interest. A simulation study showed the estimation of transmission dynamics was sensitive to the choice of prior information [27]. In addition, this model overlooked the instant-level transmission heterogeneity.

## Instant-individual heterogeneity model

For directly transmitted diseases such as SARS-CoV, MERS, Ebola, or COVD-19, the instant individual reproduction number is affected by a complex mixed factors of host, pathogen and environmental factors [1, 28]. Therefore the reproduction number is specific to time and individual. Here we assumed $v_{s,t}^i$ to be a random variable, and its values are drawn independently, for each individual $i$ and each instant $t$, from a *Gamma* distribution with mean of $w_{t-s} R_t$ and the rate of $\frac{k_t}{R_t}$, that is,

$$v_{t,s}^i \sim \Gamma(w_{t-s}k_t, \frac{k_t}{R_t}) \tag{2}$$

Under this random IIRN assumption, heterogeneous transmission stems from the variation in reproduction numbers of different individuals and at different times. And superspreading events were likely triggered by those important realizations from the right-hand tail of the distribution of IIRN, which indicated a random mixture of host, pathogen and environmental factors of assisting the rapid transmission of disease [28].

The parameter $k_t$ in (2), referred to as (instantaneous) dispersion number, was introduced to control the transmission heterogeneity. Similar to the explanation of instantaneous reproduction number $R_t$ in [26], the instantaneous dispersion number $k_t$ also controls the variation in the offspring distribution of a random infected case. Suppose the transmission dyanmics remains the same (i.e., the $R_t$ and $k_t$ keep constant) during the infectious time of the $i$-th case, its individual reproduction number over the whole infectious period is the sum of independent IIRNs over all infectious instants, that is $v_s^i = \sum_{t \geq s} v_{t,s}^i \sim \Gamma(k_t, \frac{k_t}{R_t})$. As a consequence of this

*Gamma-Poisson* mixture, the total offspring of the particular case is Negative Binomial distributed as

$$I_s^i \sim \text{NegB}(k_t, \frac{k_t}{R_t + k_t})$$

with the mean of $\mu = E(I_s^i) = R_t$ and variance $\sigma^2 = R_t (1 + R_t/k_t)$. The offspring distribution was identical to the standard model of transmission heterogeneity in [1]. Obviously, the dispersion number $k_t$ is an empirical measure of degree-of-transmission heterogeneity, with smaller $k_t$ indicates higher variance in offspring distribution (i.e., higher level of heterogeneity). When $k_t$ decreases both the likelihood of super- and that of sub-spreading events increase [22]. Traditionally, it is regarded as *significant* transmission heterogeneity when $k_t$ gets smaller than 1 [1].

Based on the random IIRN assumption, the total effect of all the infected cases on the newly infection at time $t$ was the sum of their independent IIRNs, that is, $v_t = \sum_{s \leq t,i} v_{t,s}^i \sim \Gamma(k_t \Lambda_t, \frac{k_t}{R_t})$. Furthermore, the incidence $I_t$ is Negative-Binomial distributed as

$$I_t | \bar{I}_1^{t-1} \sim \text{NegB}(k_t \Lambda_t, \frac{k_t}{R_t + k_t}), \tag{3}$$

that is,

$$P(I_t | \bar{I}_1^{t-1}, w, R_t, k_t) = \binom{\Lambda_t k_t + I_t - 1}{\Lambda_t k_t - 1} (\frac{R_t}{R_t + k_t})^{I_t} (\frac{k_t}{R_t + k_t})^{\Lambda_t k_t}, \tag{4}$$

This incidence model is referred to as the Instant-individual heterogeneity model.

If assuming that the transmission dynamics (i.e., reproduction number $R_t$ and dispersion number $k_t$) was constant, it is possible to obtained the overall estimate of both transmission heterogeneity and reproduction number simultaneously by fitting the observed incidence time series to this model. Additionally, in real epidemics, the transmission dynamics may vary with time because of changes in host and environmental factors. A common framework for monitoring the temporal trend of transmission dynamics is to assume constant transmissibility potential and heterogeneity over a time period $[t - \tau + 1, t]$, measured by $R_{t,\tau}$ and $k_{t,\tau}$ [19]. With this assumption, the likelihood of the incidence $I_{t-\tau+1}, \cdots, I_t$ given the transmission dynamics ($\{R_{t,\tau}, k_{t,\tau}\}$) and conditioned on the previous incidences $I_1, \cdots, I_{t-\tau}$ is

$$P(I_{t-\tau+1}, \cdots, I_t | \bar{I}_1^{t-\tau}, R_{t,\tau}, k_{t,\tau})$$

$$= \prod_{s=t-\tau+1}^{t} \binom{\Lambda_s k_{t,\tau} + I_s - 1}{\Lambda_s k_{t,\tau} - 1} (\frac{R_{t,\tau}}{R_{t,\tau} + k_{t,\tau}})^{I_s} (\frac{k_{t,\tau}}{R_{t,\tau} + k_{t,\tau}})^{\Lambda_s k_{t,\tau}}, \tag{5}$$

On the basis of this joint likelihood function of both reproduction number and dispersion number, it is possible to infer the real-time transmission heterogeneity from the incidence data, which gives a more complete view of the characteristics of disease spreading. In particular, the maximum likelihood estimation of the reproduction number with this new likelihood function is given by $\hat{R}_{t,\tau} = \frac{\sum_{s=t-\tau+1}^{t} I_s}{\sum_{s=t-\tau+1}^{t} \Lambda_s}$, which coincides with that of the homogeneous model [19, 29]. This property guarantes that the estimation of reproduction number with our model is robust to the bias of constant under-reporting rate (shown in Results). It is also possible to derive the posterior distribution of $R_t$ and $k_t$ by using a Bayesian framework.

## Simulation and analysis of incidence time series

We applied the Instant-individual heterogeneity (IIH) model to simulated datasets to test its accuracy under various levels of transmission heterogeneity and reproduction number. Each simulation began with 10 infected index cases. We assumed constant reproduction number $R$ and dispersion number $k$, and simulated the newly infection according to the likelihood of the incidence in (4). In particular, we calculated the total infectiousness at time $t$ as $\Lambda_t = \sum_{s=1}^{t-1} I_s w_{t-s}$, and then generated the incidence $I_t$ according to the Negative binomial distribution as in (3).

Our simulation of incidence data could be referred to as population-level simulation of incidence data. Additionally, we performed an individual-level simulation, where we firstly simulated the secondary case of each infectious case at time $t$ and then aggregated over these new cases to generate the incidence data $I_t$. In particular, we fixed the heterogeneity for all infectious cases at time $t$ as $k_t$, and drew a random reproduction number $v_{s,t}^i$ (2) for the $i$-th case infected as time $s$ ($s < t$) to represent its infectiousness at time $t$. Consequently, the particular case would generate $I_{s,t}^i \sim \text{Pois}(v_{s,t}^i) = \text{NegB}(w_{t-s}k_t, \frac{k_t}{R_t+k_t})$ new cases at time $t$. Under the branching process framework, the transmission of all index cases was regarded as independent, so the total incidence $I_t$ is the sum of a series of independent *Negative Binomial* variables, that is,

$$I_t = \sum_{s<t}\sum_i I_{s,t}^i \sim \sum_{s<t}\sum_i \text{NegB}(w_{t-s}k_t, \frac{k_t}{R_t+k_t}) = \text{NegB}(\Lambda_t k_t, \frac{k_t}{R_t+k_t}). \tag{6}$$

where the last step comes from the additivity of independent Negative Binomial variables with the same probability of success [30]. The Eq 6 means that the daily incidence $I_t$ has the same distribution under both population-level simulation and individual-level simulation. We also compared the performance of the IIH model under these two methods of simulation in the Result section.

Specifically, we set three levels of reproduction number $R$ as 1.1, 1.3 and 1.5; and four levels of dispersion number $k$ as 0.2, 0.5, 2, and 5. The serial interval distribution was set as a gamma distribution with mean of 5.2 days and the standard deviation of 1.72 days as in the COVID-19 [31]

We simulated the epidemic for 24 days and chose the incidence data from the last time window (e.g., 7 days) to perform estimation. We assumed non-informative priors of uniform distribution over $[10^{-6}, 100]$ and $[0.1, 10]$ for the reproduction number and the dispersion number respectively. Both the maximum a posteriori (MAP) estimation and the 95% highest posterior density (HPD) interval of reproduction number and dispersion number were generated. We also varied the window size used in the estimation as 7, 14 and 21 days to test the effect of data size.

The simulation was repeated 100 times under each condition. Three criteria were used to evaluate the accuracy of the estimation. Firstly, the *relative* median absolute deviations (MADs) were calculated for the estimation of $R$ and $k$ respectively, which was defined as:

$$\text{MAD} = \text{median}_i(|\hat{\theta}_i/\theta - 1|),$$

where $\theta$ is the true value of parameter, and $\hat{\theta}_i$ is the estimation of parameter based on the $i$-th simulation. The criterion of MAD was chosen because the fitting of Negative binomial distribution was unstable [1]. Secondly, the coverages of the 95% HPDs of reproduction number $R$ and $k$ were calculated. Thirdly, the probability of correctly identifying heterogeneity, namely the proportion of simulations where both the true dispersion number $k$ and its estimate were

larger or smaller than 1, was calculated for the estimation of *k*. Lastly, to measure the convergence of the posterior sampling in the calculation with Bayesian model, we calculated the effective sample size (ESS) for each estimate of parameter [32]. Usually, a large ESS (i.e., ESS > 200) stands for a good convergence of sampling.

In addition, to test the performance of the IIH model under the scenario with irregular reporting rate, we set the reporting frequency as one day and triple days iteratively. We simulated the incidence data with regular reporting rate and then aggregated the data over the irregular delay (i.e., triple days) to generate the incidence data with irregular reporting rate.

When analyzing the irregularly reported data, we divided the incidence averagely to the regular time instants (i.e., per day) to generate a synthetic regularly reported data. We analyzed this synthetic data with the IIH model to generate the estimation of reproduction number and dispersion number.

## Analyzing real epidemic data

We also applied the instant-individual heterogeneity model to disease incidence time series from several past outbreaks where the levels of heterogeneity were estimated on the basis of contact tracing data or individual level spatial information. The commonly used transmission heterogeneity model in [22] (referred to as the instant-level heterogeneity model) was also used to analyze these incidence time series under the same setting for comparison.

We retrieved the epidemic curves, as well as the mean and standard deviation of the serial intervals of these epidemics from the literature (Table 1). These epidemics were classified into two groups according to the way of estimating transmission heterogeneity in previous studies. The first group (static scenario) includes three epidemics, i.e., COVID-19 in Hongkong, China between 2020-01-24 and 2020-04-28 (referred to as COVID-19 in Hongkong, 2020), COVID-19 in Tianjing, China between 2020-01-21 and 2020-02-15 (referred to as COVID-19 in Tianjing), and MERS in several places in South Korea between 2015-05-11 and 2015-06-26 (referred to MERS in South Korea). For each of these outbreaks, previous study assumed constant

**Table 1. Description of historic epidemic data analyzed.**

| Category | Disease | Location | Duration of Outbreak | Mean (SD) serial interval $^\diamond$ | Reference for Mean (SD) | Source of Incidence time series |
|---|---|---|---|---|---|---|
| Static scenario | COVID-19 | Hongkong, China | from 2020-01-24 to 2020-04-28 | 5.2 (1.72) | [31] | [36]* |
| | COVID-19 | Tianjing, China | from 2020-01-21 to 2020-02-15 | 5.2 (1.72) | [31] | [10] |
| | MERS | South Korea | from 2015-05-11 to 2015-06-26 | 12.6 (2.8) | [37] | [38] |
| | Measles | Canada | from 2019-01-01 to 2019-08-31 | 14.5 (3.25) | [39] | [35] |
| Time Varying scenario | Ebola | Freetown, Sierra Leone | from 2014-08-04 to 2015-03-29 | 15.3 (9.3) | [40] | [40] |
| | COVID-19 | Georgia, United States | from 2020-03-01 to 2020-05-03 | 5.2 (1.72) | [31] | [41]* |
| | COVID-19 | Hongkong, China | from 2020-01-23 to 2021-04-05 | 5.2 (1.72) | [31] | [36]* |
| | COVID-19 | South Africa | from 2021-05-01 to 2022-01-09 | 5.2 (1.72) | [31] | [36]* |

*Dataset were accessed on 2022-02-01;

$^\diamond$ SD: standard deviation.

transmission parameters over the study period and estimated the overall $R$ and $k$ on the basis of contact-tracing data [10, 33, 34]. Here we followed this assumption and applied the IIH model and instant-level model to the incidence data over the same period to get the overall estimation of $k$ and $R$, which were compared with the corresponding records in literatures. We also calcuated the overall estimation of transmission transmission heterogeneity of measles epidemic in Canada during 2019 [35] and compared with the recorded heterogeneity of measles epidemic in Canada from 1998 to 2001 [1]. This comparison is not formal since the analyzed data and the recorded heterogeneity came from two epidemics.

The second group (time-varying scenario) includes three outbreaks including: the Ebola epidemic between Aug 04, 2014 (week 36), and March 29, 2015 (week 13), in the capital Freetown of Sierra Leone (referred to as Ebola, Sierra Leone); the COVID-19 epidemic in five counties (i.e., Cobb, DeKalb, Gwinnett, Fulton and Dougherty) in Georgia, United State during the period between March 1, 2020 and May 3, 2020 (referred to as COVID-19, Georgia), and the COVID-19 epimdemic in Hongkong, China from January 23, 2020 to April 5, 2021 (referred to as COVID-19, Hongkong, 20–21). For each of the epidemics, previous studies analyzed the transmission heterogeneity of several periods on the basis of individual-level spatial information as well as population density data [3, 9, 16]. To make a comparison with the recorded temporal trends of heterogeneity in literatures, we assumed constant transmission dynamics over a sliding time window to reveal the real-time estimation of $k_t$ and $R_t$ on the basis of the incidence data. We set the window length as 7 time-steps (i.e., days or weeks depending on the frequency of incidence data collection) for the first two epidemics and 14 days for the third epidemic, which was recommended in [20] when monitoring the temporal trend of reproduction number.

In addition, we also explored the transmission heterogeneity of the variant of Omicron by applying the IIH model to the incidence time series from the South Africa between 2021-05-01 and 2022-01-07. The real-time estimation of transmission dynamics was also generated as the same procedure in the time-varying scenario.

During these studies, the discrete distribution of the serial interval was then obtained by assuming a gamma distribution truncated by Mean+3*SD of serial interval. We used Bayesian Monte Carlo Markov Chain algorithm to calculated the posterior distribution from the likelihood functions of (4) and (5) by assuming non-informative priors of uniform distribution over $[10^{-6}, 100]$ and $[0.1, 10]$ for the reproduction number and the dispersion number respectively. The resulting 95% high probability domain (95% HPD) could be directly compared with those 95% confidence intervals in literature. The inference algorithm was implemented via the open-sourced python package of pymc3 [42]. All the codes for this study are available online: https://github.com/yunPKU/infer_heterogeneity_from_incidence

## Sensitivity analysis

Underreporting and misspecification of serial interval are ubiquitous biases for the analysis of epidemiological data [9, 29]. To explore the effect of these biases on the estimation of real-time dispersion number and reproduction number, we performed sensitivity analysis on the basis of the epidemic data of Ebola, Sierra Leone [40]. Firstly, we explore the effect of underreporting on our analysis. We assumed constant reporting rate through the epidemic and tested 4 reporting rates (i.e., $\rho$ = 0.8, 0.6, 0.4, 0.2). With each rate, we generated synthetic incidence time series in the Ebola epidemic by increasing the recorded incidence data proportionally.

Secondly, we considered the effect of time-varying reporting rate. As shown in [43], the reporting rate increases during the epidemic wave and reaches a plateau at the end of the epidemic. Therefore, in the Ebola epidemic, we initiated the reporting rate as 0.3, increased the

rate linearly to 1.0 at the end of the fourth period (i.e., on 2015-02-09), and remained the rate unchanged till the end (i.e., 2015-03-29). We also generated the synthetic incidence time series according to the time-varying reporting rate.

Lastly, we tested the errors in the serial interval by analyzing the Ebola epidemic data with biased serial interval distribution. We performed estimation with three values of bias for the mean (i.e., -7 days, 7 days, and 14 days) and three biases for std (i.e., -3.5 days, 3.5 days and 7 days) respectively.

## Results

### Evaluation on simulated data

In our stimation, there were on average about 128, 266, and 600 new infections during 24 days under the conditions of $R = 1.1, 1.3$, and 1.5 respectively (Fig A in S1 Text). These infection sizes were comparative with those in real datasets. For example, there were totally 135 infections in the COVID-19 outbreak of Tianjing, China in 2020 which was lasting for 25 days. Typical incidence curves are shown in Fig A in S1 Text.

Under the scenario with regular reporting, our model could accurately estimate the overall dispersion number and the reproduction number providing sufficient data (Fig 1A-C). As the window length increased (Fig 1A-C), the relative median absolute deviations (MADs) of these two estimates $k$ and $R$ showed a decreasing trend under all simulation settings. Also, the coverage of 95% HPD of $k$, the probability of identification of $k$, and the coverage of 95% HPD of $R$

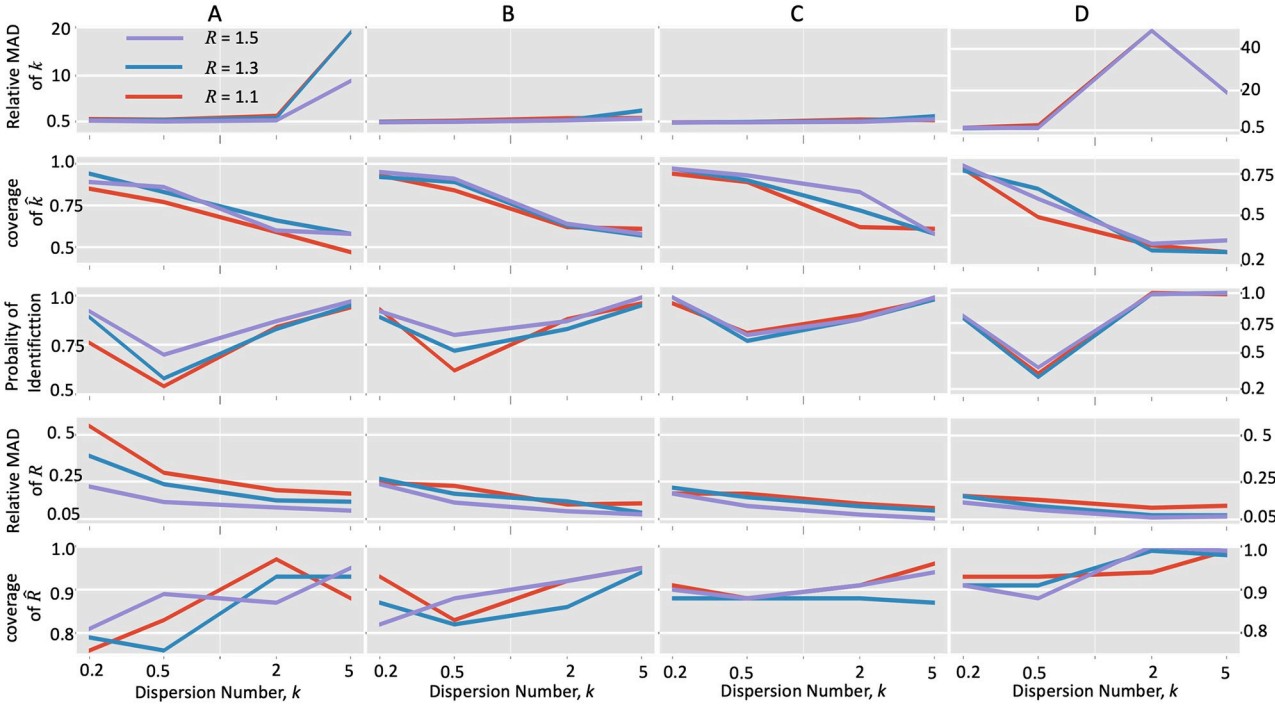

**Fig 1. Accuracy of the instant-individual heterogeneity model in estimating transmission dynamics with simulated data.** Incidence data were generated with the instant-individual heterogeneity model (4) with given reproduction number and dispersion number. Each simulation began with 10 cases and stopped at 24 days. The relative MADs and the coverage of 95% high probability density interval were calculated for the estimation of reproduction number $R$ and dispersion number $k$ respectively. The probability of identification (defined in the section of methods) was also calculated for the estimation of dispersion number $k$. A, B, and C. Estimation with daily reported incidence data of different time lengths, i.e., window size = 7 days (A), 14 days (B), and 21 days (C); D. Estimation with irregularly reported incidence data, where the incidence data were generated every day or every three days iteratively.

increased with the window length. All these findings suggested that more accurate estimation would be obtained with more data.

It should be noted that the estimation accuracy of dispersion parameter $k$ became worse at the homogenous conditions (i.e., $k > 1$). The relative MADs of $k$ grew up and the coverage of 95% HPD of $k$ fell down in each subplot as the true $k$ increased. This observation is consistent with previous simulation study of dispersion number [1, 44], which found that the dispersion number is likely to be overestimated for small sample size. However, the probabilities of identifying of $k$ under the homogeneous conditions were closer to 0.9, suggesting that our model could correctly identify this homogeneous condition. In addition, as to the estimation of $R$, the relative MAD decreases and the coverage of 95% HPD increases when the true $k$ increaseed, suggesting that the estimate of $R$ is more accurate for the homogeneous situation.

Under the scenario with irregular reporting, our estimation of dispersion number $k$ became worse with larger relative MADs, lower empirical coverage of 95% HPD and lower probability of correcting identify heterogeneity (Fig 1D). These biases came from the fact that irregular reporting would reduce the sample size given the same study period and made the rare events being harder to occur in the finite sample. Moreover, the estimation of reproduction number $R$ was less affected by the irregular reporting for lower values of $R$ (i.e., $R = 1.1$ and 1.3). The relative MAD remained low and the empirical coverage 95% HPD remained high.

All the above esitmation of parameters showed sufficient convergence in Bayesian sampling. The effective sample size (ESS) of all the estimation remained higher than 3000 (Fig B in S1 Text) in most settings. In addition, the perfomrance of the IIH model was not considerable affected (Fig C in S1 Text) under the condition with more simulation runs. We also verified the IIH model with the incidence data generated by the individual-level method of simulation. Particularly, we test the IIH method under the condition of window size = 21days. The performance of the IIH model with these new data was comparable with the above results from the population-level simulation data (Fig D in S1 Text).

## Validation with Real epidemics

**Static scenario.**   When analyzing the incidence data of three epidemics with the instant-individual heterogeneity model, our estimates of the dispersion number $k$ were 0.51 (95% HPD: 0.16∼1.55) and 0.10 (95% HPD: 0.056∼0.17) for the epidemics of COVID-19 in Tianjin and the MERS in South Korea, respectively, which were consistent with those published estimates based on contact tracing data (Fig 2A) [10, 34]. Our estimation of $k$ for the epidemic of COVID-19 in Hongkong, 2020 was 0.19 (95% HPD: 0.13 0.26), which was a little lower than the records of 0.43 (95% CI: 0.29∼0.67) of the same epidemic [33], but was comparable with the results for other epidemics of COVID-19 worldwide [45]. As to the estimation of reproduction number $R$, our estimates were consistent with previous studies (Fig 2B) for these three epidemics.

In addition, we analyzed the dispersion number of the measles outbreak in Canada between 2019-01-01 and 2019-08-31 [35]. With the IIH model, we estimated the dispersion number $k$ as 0.94 (95% HPD: 0.22∼7.47), which is overlapped with the recorded result on $k$ (i.e.,90% CI 0.12∼0.65) in Canada from 1998 to 2001 [1].

Applying the instant-level heterogeneity model to these epidemics [22], the estimates of $k$ were 2.08 (95% HPD: 0.72∼6.12) and 1.51 (95% HPD: 0.99∼2.19) for the epidemics of COVID-19 in Tianjin and in HongKong respectively, which were far from the published results (Fig 2A). For the epidemic of MERS in South Korea, the estimate of k was 0.44 (95%

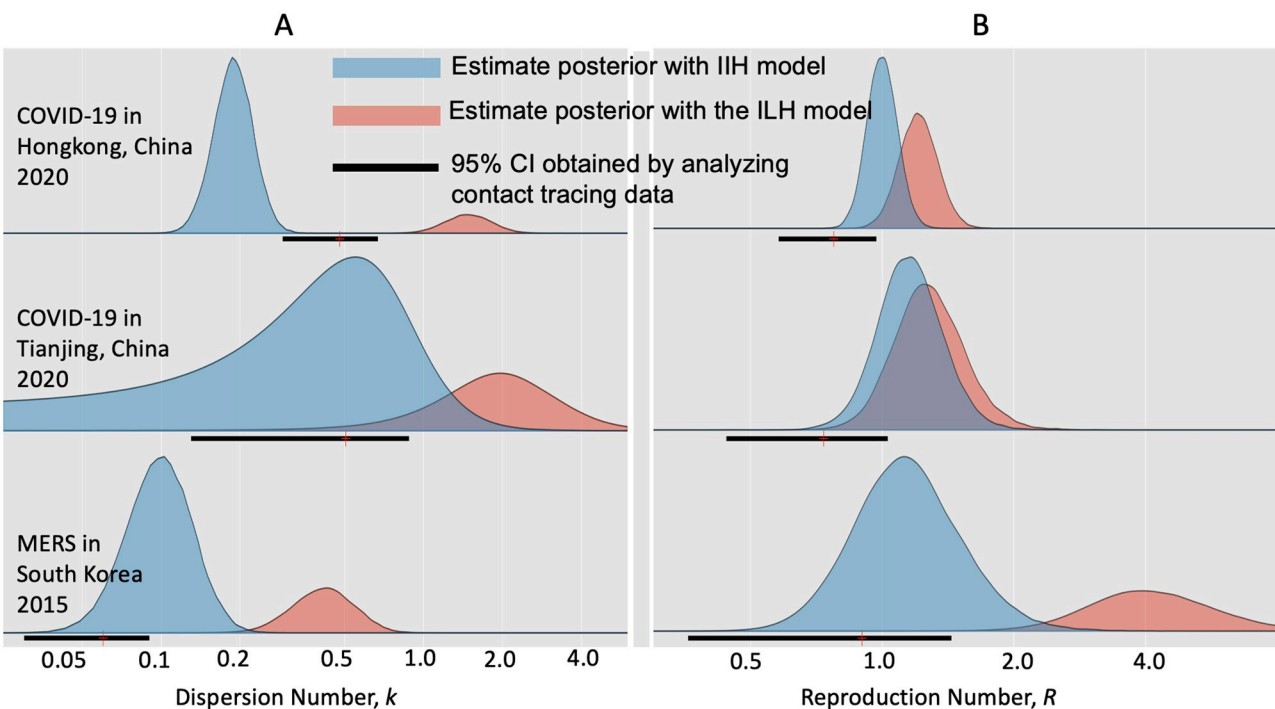

**Fig 2. Comparison of estimating transmission dynamics of three epidemics with the instant-individual heterogeneity (IIH) model and the instant-level heterogeneity (ILH) model in [22].** During each epidemic, transmission dynamics (i.e., reproduction number $R$ and dispersion number $k$) were assumed constant. Colored areas showed the posteriors of the estimates by analyzing incidence times series. Black solid lines represented the estimates in literatures obtained by analyzing the contact tracing data of these epidemics [10, 33, 34]. A: Estimation of reproduction number ($R$); B: Estimation of dispersion number ($k$).

HPD: 0.25∼0.69), being close to the published estimates, but the estimate of $R$ was 4.04 (95% HPD: 2.13∼8.07), which was far from the records [34].

We summarized the overall estimation of dispersion number with the IIH model for seven epidemics involved in this study and compared with published results (Table A in S1 Text). Except for the epidemic of COVID-19 in Hongkong 2020, our estimates were consistent with published estimates. As to the exceptional example of COVID-19 in Hongkong 2020, our estimates, being 0.19, also fell within the reference range of the heterogeneity of SARS-CoV-2 [45–47].

**Time-varying scenario.** By assuming that the transmission parameters remanin constant over a time window 7 steps (i.e., days or weeks depending on the frequency of incidence data collection), we obtained the real-time estimation of the dispersion number ($k_t$) as well as the reproduction number ($R_t$) over an epidemic. Firstly, we analyzed the weekly incidence of probable and confirmed cases of Ebola between August 4th, 2014, and March 29th, 2015, in the capital Freetown of Sierra Leone. By setting the reference time of 2014-11-01 as in [16], the whole duration was divided into 5 periods (P1 to P5, Fig 3).

We estimated the overall dispersion number over the whole period to 0.065 (95% HPD: 0.037∼0.11), which slightly overlapped with the published results of 0.18 (95% CI: 0.10∼0.26) of $k$ from the Ebola epidemic in Guinea, 2014 [11]. As to the real-time dispersion number ($k_t$) for the epidemic, our estimate remained stable during the first two periods (P1 and P2) and decreased since the third period and then reached the lowest level around 0.1 in the forth period. At last, the $k_t$ bounced up to around 0.2 in the last period (Fig 3C). This temporal trend of $k_t$ was consistent with previous study based on individual level spatial information,

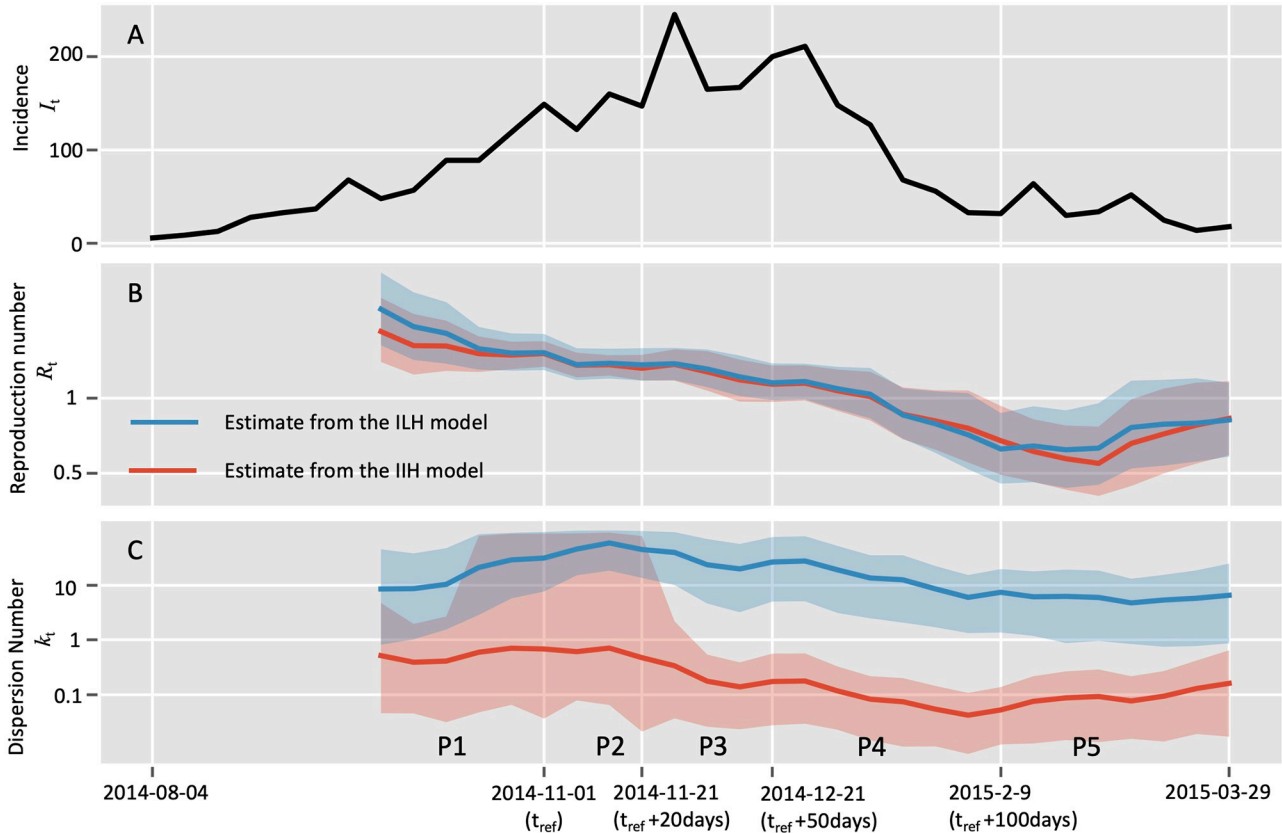

**Fig 3. Comparison of estimating real-time transmission dynamics of the Ebola epidemic between Aug 04, 2014 (week 36), and March 29, 2015 (week 13), in the capital Freetown of Sierra Leone.** Transmission dynamics (i.e., reproduction number $R$ and dispersion number $k$) were assumed constant over a window of 7 weeks, and the estimates were obtained by analyzing the incidence data of the time window. Solid lines show the mean estimates from two methods. Red curves and blue curves represent the estimation from the instant-individual heterogeneity model (IIH) and the instant-level heterogeneity (ILH) model respectively. The shaded areas show the 95% high probability density (HPD) intervals. As in [16], the reference time $t_{ref}$ was set as 2014-11-01, and the whole time period was divided into five periods as: from 2014-10-20 to $t_{ref}$ (period 1), $t_{ref}$ to $t_{ref}$ +20 days (period 2), $t_{ref}$ +20 days to $t_{ref}$ +50 days (period 3), $t_{ref}$ + 50 days to $t_{ref}$ + 100 days (period 4), and thereafter (period 5).A: Incidence data of the confirmed and probable cases; B: Estimation of reproduction number ($R_t$); C: Estimation of dispersion number ($k_t$).

suggesting the transmission heterogeneity were becoming more significant as the epidemic went on and might be crucial to driving the spreading of Ebola disease in the study area [16]. In contrast, the instant-level model generated much higher estimate of dispersion number $k_t$ which remained above 1, suggesting it failed to reveal the significant transmission heterogeneity during this outbreak (Fig 3C).

We also noted that both the IIH model and the instant-level model gave similar estimation of the real-time reproduction number, which showed a declining trend in most part of the period, and was below 1 since the middle of the fourth period (Fig 3B).

Secondly, we validated the IIH model with the COVID-19 incidence data, between March 1, 2020 and May 3, 2020, in five counties of Georgia state, USA (Fig 4). We estimated the overall dispersion number based on the aggregated incidence data over the five counties as 0.023 (95% HPD: 0.016~0.031) which was consistent with the recorded estimate 0.009 (95% CI: 0.007~0.348) on the basis of individual spatial information from the same epidemic [45].

For the real-time dispersion number ($k_t$), our estimates for all counties remained lower than 1 in most days (Fig 4C), suggesting significant transmission heterogeneity of COVID-19 in all these counties [9]. Notably, the transmission heterogeneity became mostly significant in

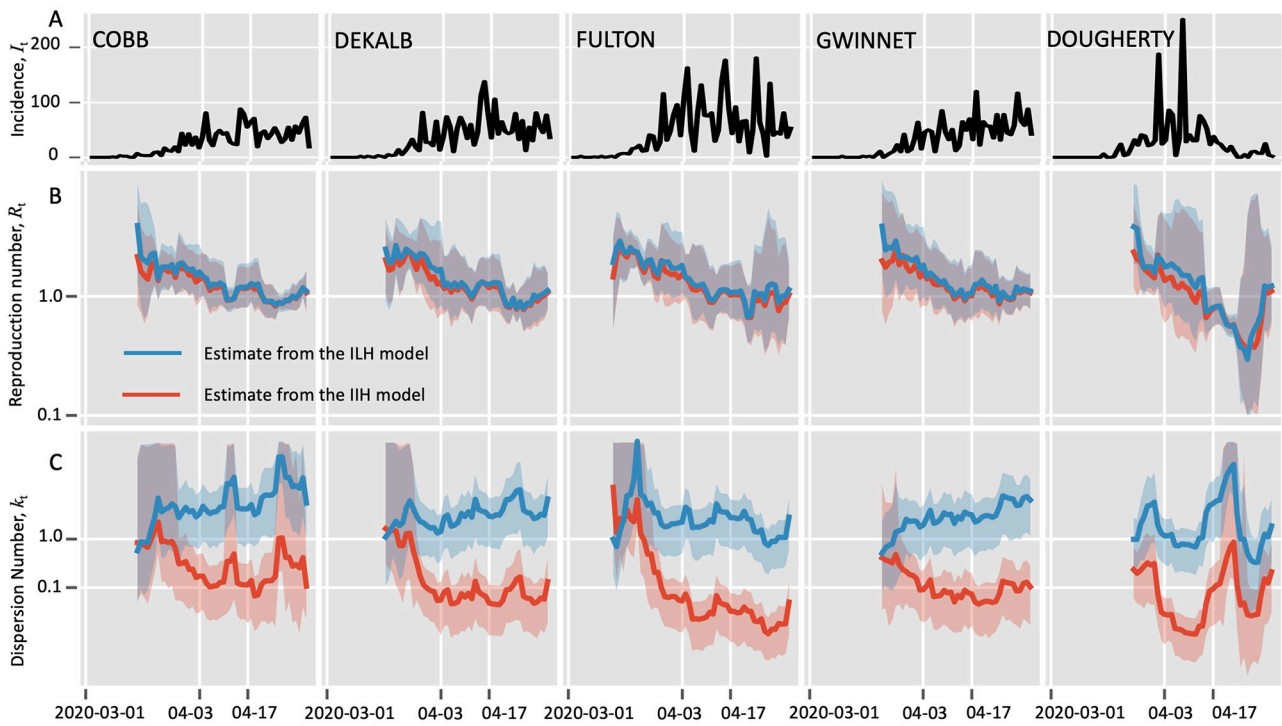

**Fig 4. Comparison of estimating real-time transmission dynamics of the COVID-19 epidemic between March 1, 2020 and May 3, 2020, in five counties of Georgia state, USA.** Transmission dynamics (i.e., reproduction number $R$ and dispersion number $k$) were assumed constant over a window of 7 days, and the estimates were obtained by analyzing the incidence data of the time window. Solid lines show the mean estimates from two methods, i.e., red curves and blue curves represent the estimation from the instant-individual heterogeneity model (IIH) and the instant-level heterogeneity (ILH) model respectively. The shaded areas show the 95% high probability density (HPD) intervals. As in [9], the reference time was set as April 3rd, 2021 when the shelter-in-place order was announced. The whole study period was divided into three periods, i.e., before April 3rd, between April 3rd and April 17th, after April 17th. A: Incidence data of the confirmed and probable cases; B: Estimation of reproduction number ($R_t$); C: Estimation of dispersion number ($k_t$).

the rural area (Dougherty) with the estimated $k_t$ reached the lowest level of around 0.01 in the second period, which was consistent with the documented superspreading event in this county [48]. In contrast, the instant-level model gave the overall estimation of dispersion number as 3.08 (95% HPD: 1.97~4.46) and generated the real-time estimation of $k_t$ being above 1 for each county, which failed to identify the significant transmission heterogeneity in all these counties.

The IIH model and the instant-level model gave similar estimation of reproduction number $R_t$ (Fig 4B). We found that the reproduction numbers in four countries (i.e., except for Gwinnet) declined below 1 short after Apr-17 (i.e., 2 weeks after the shelter-in-place order), suggesting the order was effective to reduce the transmission of COVID-19. Similar to the findings in [9], our IIH model also indicated that the urban area of Dougherty was the first country where $R_t$ declined below 1.

Lastly, we validated the proposed IIH model with the COVID-19 incidence data, between Jan 23, 2020 and Apri 5, 2021, in Hongkong, China (Fig 5) which consisted of three epidemic waves of COVID-19 transmission. The overall dispersion number of the whole study period was estimated to be 0.16 (95% HPD: 0.14~0.19), which overlapped with the published estimate of 0.20 (95% CI: 0.16~0.25) based on contact tracing data [49]. We set the window length as 14 days to get a smoothing estimate of the temporal trend of transmission pattern. With the IIH model, we found that the real-time dispersion number $k_t$ fluctuating between 0.1 and 1

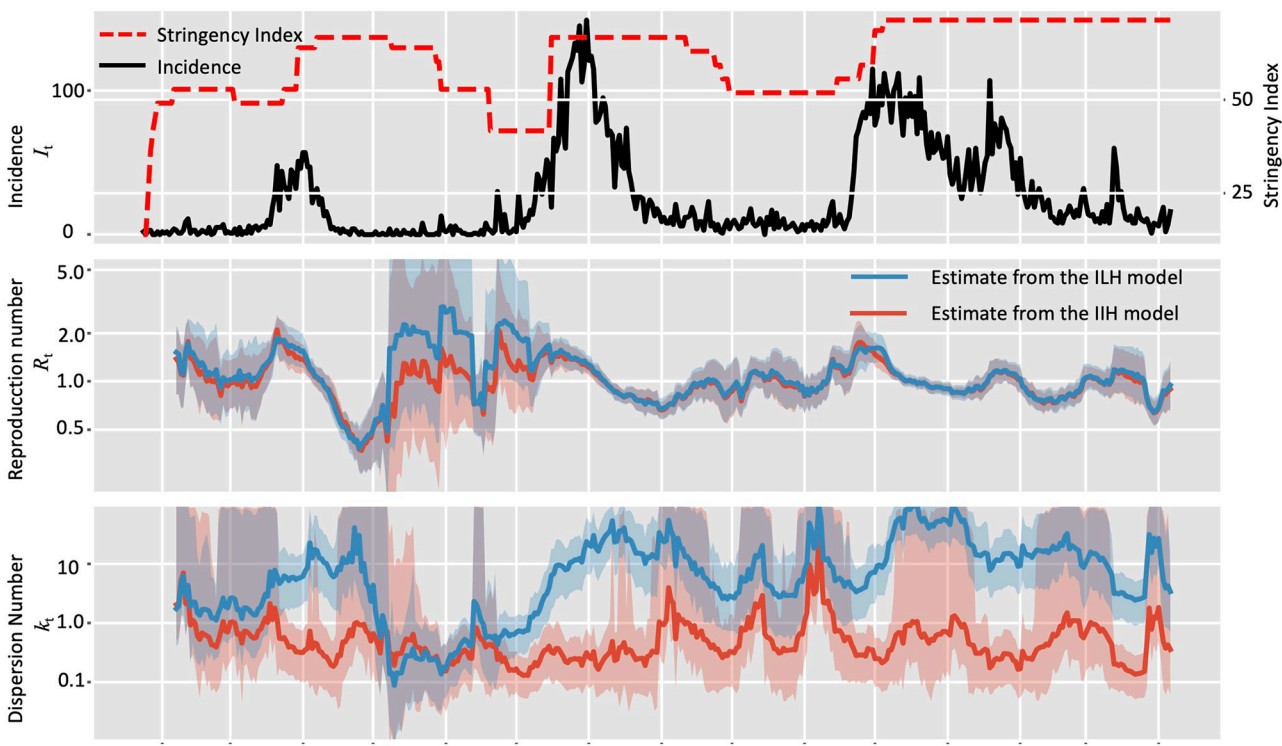

**Fig 5. Comparison of estimating real-time transmission dynamics of the COVID-19 epidemic between Jan 23, 2020 and Apri 5, 2021, in Hongkong, China.** Transmission dynamics (i.e., reproduction number R and dispersion number k) were assumed constant over a window of 14 days, and the estimates were obtained by analyzing the incidence data of the time window. Solid lines show the mean estimates from two methods. Red curves and blue curves represent the estimation from the instant-individual heterogeneity model (IIH) and the instant-level heterogeneity (ILH) model respectively. The shaded areas show the 95% high probability density (HPD) intervals. A: Incidence data of the confirmed cases and government stringency data in South Africa; B: Estimation of reproduction number ($R_t$); C: Estimation of dispersion number ($k_t$).

during the study period (Fig 5C), which was consistent with the results on the basis of contact tracing data [49]. We also found a significant negative relationship between the dispersion number $k_t$ and the NPI stringency with the slope as -0.02 (95% HPD: -0.036 ∼-0.0044), which was also consistent with the published estimation of -0.03 (95% CI: -0.04 ∼-0.02) [49]. In contrast, the real-time dispersion number from the ILH model was fluctuating mostly above 1, indicating the method failed to recognize the significant heterogeneity in these epidemic waves. This example validated that the IIH model could reveal the temporal trends of $k_t$ over multiple epidemic waves.

**Sensitivity analysis.** By analyzing the synthetic data with the IIH model, we found that the real-time dispersion number ($k_t$) decreased as the reporting rate decreased, suggesting that the estimation of heterogeneity was conservative if there were a lot of missing cases. This effect of reporting rate was not considerable even when the reporting rate decreased to 0.4 (i.e., 60% cases were missing), where the estimation of $k_t$ was still covered by the 95% HPD obtained under the 100% reporting rate (Fig 6B).

Also, we found that the real-time $k_t$ at different times reduce by almost the same proportion as the reporting rate decreased. Particularly, the $k_t$ reduced on average by 22%, 45%, 64%, and 83% as the reporting rate decreased to 0.8,0.6,0.4, and 0.2 respectively, resulting in similar temporal trends of $k_t$ under different reporting rates. It was interesting that the estimate of $k_t$ under time-varying reporting rate (dashed line in the Fig 6B) crossed these curves of $k_t$ as the

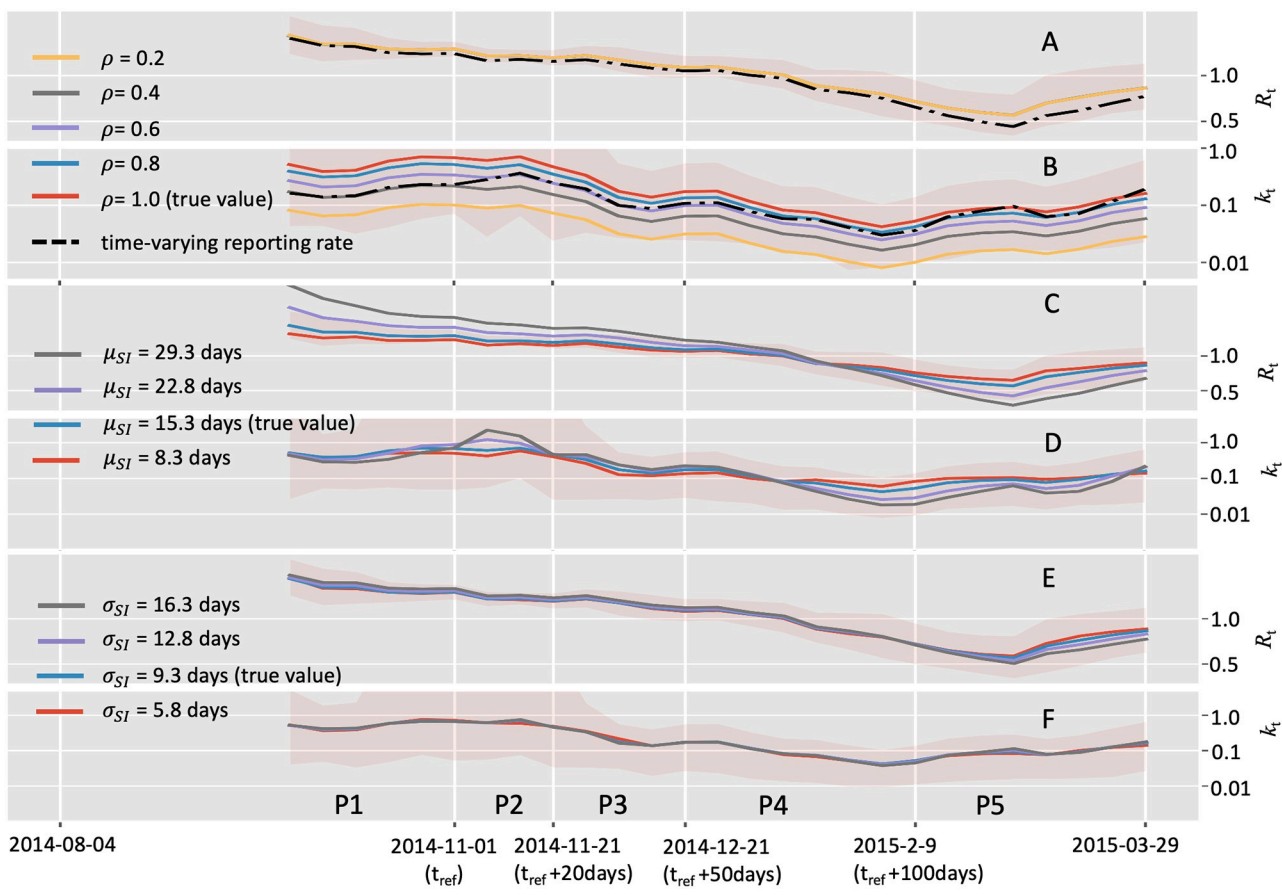

**Fig 6. Effects of underreporting rates and misspecification of the serial interval on estimating transmission dynamics with the instant-individual heterogeneity model.** Synthetic data incorporating missing cases were generated on the basis of the incidence data from the Ebola epidemic between Aug 04, 2014 (week 36), and March 29, 2015 (week 13), in the capital Freetown of Sierra Leone. Colored lines show the mean estimates and the shaded areas show the 95% high probability density intervals under the true values. A and B: Estimation under different reporting rates ($\rho$). Dashed lines represent the estimates under the scenario with time-varying reporting rate. C and D: Estimation from different specification of the serial interval mean ($\mu_{SI}$); E and F: Estimation from different specification of the serial interval standard deviation ($\sigma_{SI}$).

reporting rate increased and kept almost similar temporal trend as others since the difference between different $k_t$ curves was small.

In addition, we found that the estimation of $R_t$ with the IIH model was unaffected by the constant reporting rate (Fig 6A). The underlying reason is that the maximum likelihood of $R_t$ under our model is identical to that of the homogeneous transmission model [19], so the estimation of $R_t$ was robust to missing cases. Even under the scenario with time-varying reporting rate, the estimation of $R_t$ (dashed line in the Fig 6A) was affected a little but not significant as the changes were covered by the 95%HPD under the ideal condition.

It has been reported that the misspecification of the serial interval (or generation interval) is a large potential source of bias when estimating reproduction number from observed incidence data [29]. We also observed considerable changes in the estimation of $R_t$ due to differences in the mean serial interval (Fig 6C). Particularly, when shortening the serial interval, the estimate of $R_t$ became lower when the true value was high (e.g., P1,P2, and P3) and became higher when the true value was low (e.g., P5). The estimation of $R_t$ was less affected by the standard deviation of serial interval (Fig 6E).

Fortunately, we found that estimation of the dispersion number $k_t$ was robust to the biases either in the mean ($\mu_{SI}$) or the standard deviation ($\sigma_{SI}$) of the serial interval (Fig 6D and 6F). All the effects were small and were covered by the 95% HPDs under the true values.

**Estimating real-time transmission heterogeneity of Omicron.**    To get a timely estimate of the transmission heterogeneity of Omicron, we applied the IIH model to the incidence data in South Africa between 2021-05-01 and 2022-01-07 [36] (acessed on 2022-02-01). This duration includes the third wave of COVID-19 caused by the Delta variant from May 2021 to September 2021, and the early stage of the potential wave caused by Omicron. With this incidence data, we could not only reveal the transmission heterogeneity of Omicron, but also we made a comparison with that of Delta.

During the period of 2021-12-01 to 2022-01-07 (referred to as Omicron wave), we estimated the overall estimation of reproduction number and the dispersion parameter were 0.97 (95% HPD: 0.82~1.16) and $2.62 * 10^{-4}$ (95% HPD: $1..58 * 10^{-4} \sim 3.94 * 10^{-4}$) respectively. To make a comparison, we focused on the epidemic wave caused by the Delta variant between 2021-06-01 to 2021-08-01 (referred to as Delta wave) during which the epidemic also experienced growth and declining. The overall estimation of reproduction number and the dispersion parameter were 1.05 (95% HPD: 0.97~1.15) and $7.14 * 10^{-4}$ (95% HPD: $4.75 * 10^{-4} \sim 1.0 * 10^{-3}$) respectively. Notably that the overall dispersion number in the Omicron wave was lower than that in the Delta wave.

By setting the window size of 7 days, we got the real-time estimation of transmission dynamics during these two periods (Fig 7). During the Omicron wave, the estimated

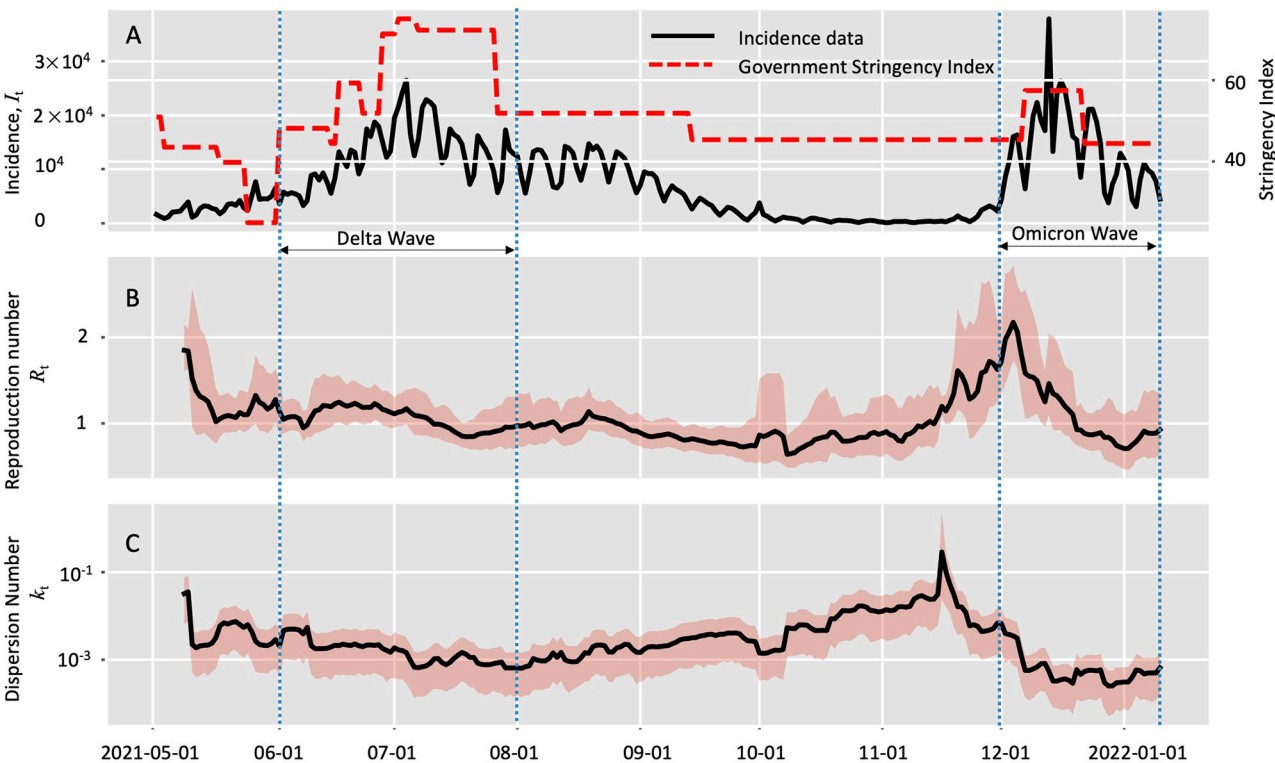

**Fig 7. Estimation of real-time transmission dynamics of the COVID-19 epidemic between 2021-05-01 and 2022-01-09 in South Africa.**
Transmission dynamics (i.e., reproduction number $R$ and dispersion number $k$) were assumed constant over a window of 7 days, and the estimates were obtained by analyzing the incidence data of the time window. Solid lines show the mean estimates and the shaded areas show the 95% high probability density (HPD) intervals. A: Incidence data of the confirmed cases and government stringency data in South Africa; B: Estimation of reproduction number ($R_t$); C: Estimation of dispersion number ($k_t$).

reproduction number $R_t$ reached the peak value of 2.10 on 2021-12-03 and then declined to the level around 0.9 after 2021-12-15. This decrease in $R_t$ might due to the deploying of control measures by the South Africa government as indicated by the highly increased government stringency index [50]. We also noted that the estimated dispersion number $k_t$ declined since 2021-12-01 and reached a stable level about $3 * 10^{-4}$ in the middle of Dec 2021.

During the Delta wave, however, we estimated reproduction number $R_t$ remained around 1 during this period which was smaller than the amount in the early of Dec 2021. In addition, the estimated dispersion number $k_t$ remained close to $10^{-3}$, which was higher than the stable level in the end of Dec 2021. Therefore, the overall and real-time estimation of transmission dynamics of these two period hint us that Omicron might not only have higher transmissibility but also a greater potential for superspreading.

## Discussion

In this study, we proposed a reliable, flexible and generic model to estimate real-time heterogeneity using incidence time series. When it was applied to the epidemic of Ebola in Sierra Leone, the epidemic of COVID-19 in the state of Georgia, USA,and the epidemic of COVID-19 in Hongkong, China 20–21, the series of daily/weekly heterogeneities, according to its estimation, paralleled with the trends reported by previous studies based on individual spatial data [3, 9].

The model successfully estimated the overall heterogeneity of six epidemics of different diseases. For example, the overall heterogeneity (in terms of dispersion number $k$) were estimated to be 0.10 (95% HPD: 0.056∼0.17), 0.065 (95% HPD: 0.037∼0.11), and 0.16 (95% HPD: 0.14∼0.19) for MERS epidemic in South Korea, Ebola epidemic in Sierra Leone, and COVID-19 epidemic in Hongkong 20–21,respectively, which were all consistent with the published estimates of the same epidemics based on contact-tracing data [11, 33, 34]. In addition, we estimated the overall heterogeneity as 0.94 (95% HPD: 0.26∼5.6) for the Measles epidemic in Canada, 2019, which also overlapped with the records of measles outbreak in Canada, 1998∼2001 [1].

Transmission heterogeneity is a ubiquitous feature in the spread of infectious disease due to a mixture of factors involving host, pathogen and environment. Accurate estimating real-time heterogeneity is vital for prediction of future epidemics and exploring targeted interventions. Existing methods of inferring transmission heterogeneity rely heavily on sophisticated data to reconstruct the offspring distribution and largely ignore the temporal change in heterogeneity. One existing model, which involves instant-level heterogeneity [22, 25], could only allow for part of the variation and hence failed to reveal accurate real-time heterogeneity. As evidenced in our analysis of the instant-level heterogeneity model, its estimation of transmission heterogeneity (in terms of dispersion number $k$) of all the real epidemics remained above the threshold of 1, indicating no significant heterogeneity in these epidemics, which completely deviated from the records in literature. Our model, however, addressed the heterogeneity with a flexible and generic way to estimate the real-time heterogeneity on the basis of incidence data, which is easy to implement and was proved reliable.

The benefits of our model stem from the two theoretical advantages. Firstly, we introduced the assumption of random instant-individual reproduction number to characterize the variation of infectiousness between different people and at different times. Both these variations were important source of the heterogeneity in transmission and therefore should be characterized in the model. This assumption is applicable for directly transmitted disease such as SARS-CoV, MERS, Ebola, and COVD-19, where the infectiousness of a particular individual at a particular instant was determined by the properties of the host and pathogen and

environmental circumstances [1, 28]. Secondly, our model is easy to implement as it employs only incidence data. We deduced the joint likelihood function of incidence data on both the reproduction number ($R_t$) and transmission heterogeneity ($k_t$), which enabled us easily to monitor these epidemiological parameters simultaneously.

When comparing the precision of different methods, we found that our estimation was less precise with broader credible intervals than the results based on contact-tracing data for the two outbreaks (i.e., MERS in South Korea 2015 and COVID-19 in Tianjin China, 2020) with smaller size (i.e., 100 200 cases). For the outbreak of COVID-19 in HongKong with more than 1,000 cases, our estimation had better precision than the result from contact-tracing data in terms of narrower credible interval. This might be related with the sample size of the outbreak, and our model might be more applicable to larger size epidemics.

This merit of our model could allow for fast and timely epidemiological surveillance, possibly even for the new SARS-CoV-2 variant of Omicron, which has been spreading wildly across the world since its first detection in November 2021 in Gauteng Province, South Africa. We estimated that the dispersion number for the epidemic wave caused by the Omicron variant (in December 2021) in South Africa was smaller than that of the Delta wave (from June to August 2021) in this country, suggesting more significant heterogeneity of Omicron. This finding was consistent with the recent study on the epidemic in South Korea with much fewer infections [51], which might be a complementary explanation to the unprecedentedly fast spreading of Omicron. Our results also highlighted the need of taking more efficient measure of to reduce people gathering and the possible superspreading events [28, 52].

During the implementation of our model, the serial interval distribution is required to approximate the infectiousness profile $w_s$. This distribution information may not be correctly obtained at the early stage of newly emerging infectious disease or may be biased for some pathogens where infectiousness occurs before symptoms. Fortunately, our model performed robust to the misspecification of serial interval (showed in results). Additionally, we could also relieve this dependence by integrating detailed epidemiological linkage data to estimate the serial interval separately [20] or extending the inference framework to incorporating estimation of serial interval distribution and transmission dynamics simultaneously as in [53].

When interpreting the results, we regarded the transmission heterogeneity estimated based on the incidence of confirmed cases accumulating over a time window till time $t$ as the result at that time. Since the confirmation of a case occur after the time of its infection, together with the delay due to the accumulation of data, our estimation of transmission heterogeneity definitely fell behind the reality. This delay might make our estimation misleading if the underlying transmission dynamics change rapidly during the period. There are two possible solutions to reduce the delay in the future. One option is to reconstruct the infection curve from the incidence data by accounting for the possible delay between infection and diagnosis as in [21, 54]. By applying our model to the transformed data, it is possible to get a more accurate estimation of the real-time transmission dynamics. Another option is to optimize the time length of data accumulation size to get a timely estimation, which could be done on the basis of certain performance constrain such as short-term predictive accuracy as in [55].

In this study, we utilize the *Gamma* distribution to characterize the transmission heterogeneity, which has been widely used in other studies. It also should be noted that the *Gamma* distribution is not suitable for all types of heterogeneity in the transmission. For example, the ongoing vaccination could incur heterogeneity as some people are vaccinated and others are not. This type of heterogeneity should play an important role especially when modelling the transmission heterogeneity in the pandemic of COVID-19, which should probably be Bimodal-distributed instead of Gamma distributed. Moreover, we picked very wide "non-informative" priors for the parameters of interesting for the direct comparison with recorded

estimation. In the future development of our method, one part is clearly to use some informative priors [56] to improve the estimation accuracy.

As shown in the simulation study, our method was likely to overestimate the dispersion number (i.e., underestimate the heterogeneity) under the homogeneous condition (i.e., the true dispersion number is larger than 1). This positive bias in k might occur because smaller samples are less likely to include rare extreme values [1, 44]. This also helps to explain why the estimate of k became worse under the condition with irregular reporting where the sample size became even smaller. Fortunately, our estimation performed better when the true value of k was smaller than 1 (i.e., heterogeneous condition). This hetergeoenous condition is more relevant in epidemiological practice than the homogeneous condition, so our method is of considerable implication in real applications. In addition, it is possible to extend the simulation study with an individual-level S-I-R type model or others being more representative of the real transmission process, which may provide stronger evidence of validating the new model. Similar studies have been done with the real-time estimation of reproduction number [29]. But the definition of real-time transmission heterogeneity under these complex models still needs further exploration.

Our model relies mainly on the incidence data which represent the overall infection aggregated over a study area. Hence the spatial heterogeneity within local scales was largely overlooked by our model. Earlier studies [57, 58] indicated that the heterogeneity in population distribution and the related social network structure may significantly affect the timing and severity of local epidemics, leading to considerable difference between the local and overall infection curves. Therefore, we should be cautious about the scope of using our estimation results, especially when designing and deploying some control measures. For example, our estimation of the transmission heterogeneity of the Omicron variant were based on the incidence data aggregated over the whole nation of South Africa. Hence our estimation could not be directly used to other epidemics of Omicron because the transmission dynamics and the correspoding incidence curve may vary with context. In addition, our method also provided a new tool which helps not only to obtain more accurate estimation of transmission heterogeneity of Omicron or other variants under different conditions, but also to reveal how various geographic, socioeconomic, and cultural environments affect the transmission dynamics of disease at different scales if more detailed data are available [58].

In summary, we proposed a simple and generic model to estimate the real-time transmission heterogeneity based on incidence data. This model could help epidemiologists better understand the complex mechanism in disease spreading, especially for those that are lack of more detailed data.

## Supporting information

**S1 Text. Further details on the simulation study and analysis of real epidemic datasets.** (PDF)

## Acknowledgments

The authors are grateful to Professor Jantien Backer for helpful discussion about the dataset of Ebola epidemic, and Ms. Jian Zhang for her careful examination of the manuscript.

## Author Contributions

**Conceptualization:** Yunjun Zhang, Tom Britton.

**Data curation:** Yunjun Zhang.

**Formal analysis:** Yunjun Zhang.

**Funding acquisition:** Xiaohua Zhou.

**Investigation:** Yunjun Zhang.

**Methodology:** Yunjun Zhang, Tom Britton, Xiaohua Zhou.

**Software:** Yunjun Zhang.

**Supervision:** Tom Britton, Xiaohua Zhou.

**Validation:** Yunjun Zhang.

**Visualization:** Yunjun Zhang.

**Writing – original draft:** Yunjun Zhang, Tom Britton.

**Writing – review & editing:** Yunjun Zhang, Tom Britton, Xiaohua Zhou.

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
