## [Decision Letter · Decision Letter 0]

31 May 2022

Dear Dr. Zhang,

Thank you very much for submitting your manuscript "Monitoring real-time transmission heterogeneity from Incidence data" for consideration at PLOS Computational Biology.

As with all papers reviewed by the journal, your manuscript was reviewed by members of the editorial board and by several independent reviewers. In light of the reviews (below this email), we would like to invite the resubmission of a significantly-revised version that takes into account the reviewers' comments.

I agree with both reviewers that this is a potentially interesting approach and covers a very timely topic in the field of computational epi, i.e., the estimation of the degree of overdispersion in secondary cases during an epidemic. However, both reviewers identified a number of substantive points that must be thoroughly addressed during the revision process. In particular, I suggest that the authors pay close attention to R1's detailed comments. Please also note that both reviewers highlighted the need for expanded sensitivity analyses and simulations to demonstrate the validity of the approach. Addressing the need for improved sensitivity analyses and simulation modeling will be a critical part of successfully revising. In addition, R1 noted that many of the estimates of the over-dispersion parameter, "k", deviate from what appears to be accepted in the literature. This is not necessarily a problem, but warrants quite a bit more explanation/justification. Beyond the reviewer comments, I might also suggest including an analysis of at least one other pathogen where the over-dispersion parameter has been widely estimated. For example, influenza.

We cannot make any decision about publication until we have seen the revised manuscript and your response to the reviewers' comments. Your revised manuscript is also likely to be sent to reviewers for further evaluation.

Sincerely,

Samuel V. Scarpino

Associate Editor

PLOS Computational Biology

Virginia Pitzer

Deputy Editor-in-Chief

PLOS Computational Biology

I agree with both reviewers that this is a potentially interesting approach and covers a very timely topic in the field of computational epi, i.e., the estimation of the degree of overdispersion in secondary cases during an epidemic. However, both reviewers identified a number of substantive points that must be thoroughly addressed during the revision process. In particular, I suggest that the authors pay close attention to R1's detailed comments. Please also note that both reviewers highlighted the need for expanded sensitivity analyses and simulations to demonstrate the validity of the approach. Addressing the need for improved sensitivity analyses and simulation modeling will be a critical part of successfully revising. In addition, R1 noted that many of the estimates of the over-dispersion parameter, "k", deviate from what appears to be accepted in the literature. This is not necessarily a problem, but warrants quite a bit more explanation/justification. Beyond the reviewer comments, I might also suggest including an analysis of at least one other pathogen where the over-dispersion parameter has been widely estimated. For example, influenza.

Reviewer's Responses to Questions

**Comments to the Authors:**

Reviewer #1: The review is uploaded as an attachment.

Reviewer #2: Attached.

**Have the authors made all data and (if applicable) computational code underlying the findings in their manuscript fully available?**

Reviewer #1: Yes

Reviewer #2: Yes

PLOS authors have the option to publish the peer review history of their article (what does this mean?). If published, this will include your full peer review and any attached files.

Reviewer #1: No

Reviewer #2: No
---

## [Decision Letter · Decision Letter 1]

6 Sep 2022

Dear Dr. Zhang,

Thank you very much for submitting your manuscript "Monitoring real-time transmission heterogeneity from Incidence data" for consideration at PLOS Computational Biology.

As with all papers reviewed by the journal, your manuscript was reviewed by members of the editorial board and by several independent reviewers. In light of the reviews (below this email), we would like to invite the resubmission of a significantly-revised version that takes into account the reviewers' comments.

Given the continued discrepancy between published estimates of "k" and those presented in this paper, I agree with R1's updated assessment and R2's prior assessment that individual-level simulations with known "k" (or at least comparable measure of super-spreading) are a necessity for this work. Without such individual-level simulations, it's simply not possible to evaluate the methodology nor is it easy to interpret the substantial differences in "k" parameters estimated in this paper vs. those reported in prior works. While I appreciate the authors' attention to the past round of reviews and verbal explanation for what may be driving the difference in "k" parameter estimates, more quantitative work is needed to both demonstrate the validity of the approach and mechanistically account for the reported discrepancies.

We cannot make any decision about publication until we have seen the revised manuscript and your response to the reviewers' comments. Your revised manuscript is also likely to be sent to reviewers for further evaluation.

Sincerely,

Samuel V. Scarpino

Academic Editor

PLOS Computational Biology

Virginia Pitzer

Section Editor

PLOS Computational Biology

Given the continued discrepancy between published estimates of "k" and those presented in this paper, I agree with R1's updated assessment and R2's prior assessment that individual-level simulations with known "k" (or at least comparable measure of super-spreading) are a necessity for this work. Without such individual-level simulations, it's simply not possible to evaluate the methodology nor is it easy to interpret the substantial differences in "k" parameters estimated in this paper vs. those reported in prior works. While I appreciate the authors' attention to the past round of reviews and verbal explanation for what may be driving the difference in "k" parameter estimates, more quantitative work is needed to both demonstrate the validity of the approach and mechanistically account for the reported discrepancies.

Reviewer's Responses to Questions

**Comments to the Authors:**

Reviewer #1: the review is uploaded as an attachment

Reviewer #2: The authors have made the caveats of their method more clearly in the revised manuscript.

**Have the authors made all data and (if applicable) computational code underlying the findings in their manuscript fully available?**

Reviewer #1: Yes

Reviewer #2: None

PLOS authors have the option to publish the peer review history of their article (what does this mean?). If published, this will include your full peer review and any attached files.

Reviewer #1: No

Reviewer #2: No
---

## [Decision Letter · Decision Letter 2]

16 Nov 2022

Dear Dr. Zhang,

We are pleased to inform you that your manuscript 'Monitoring real-time transmission heterogeneity from Incidence data' has been provisionally accepted for publication in PLOS Computational Biology.

Best regards,

Samuel V. Scarpino

Academic Editor

PLOS Computational Biology

Virginia Pitzer

Section Editor

PLOS Computational Biology

Reviewer's Responses to Questions

**Comments to the Authors:**

Reviewer #1: I thank the authors for their careful and thorough responses. I believe the manuscript has substantially improved and my stated concerns have been addressed.

**Have the authors made all data and (if applicable) computational code underlying the findings in their manuscript fully available?**

Reviewer #1: Yes

PLOS authors have the option to publish the peer review history of their article (what does this mean?). If published, this will include your full peer review and any attached files.

Reviewer #1: No

---

## [Editor Report · Acceptance letter]

27 Nov 2022

PCOMPBIOL-D-22-00546R2 

Monitoring real-time transmission heterogeneity from Incidence data

Dear Dr Zhang,

I am pleased to inform you that your manuscript has been formally accepted for publication in PLOS Computational Biology. Your manuscript is now with our production department and you will be notified of the publication date in due course.

With kind regards,

Olena Szabo
